# Format-dependent and format-independent representation of sequential and simultaneous numerosity in the crow endbrain

Helen M. Ditz[1] & Andreas Nieder[1]*

Humans' symbolic counting skills are built on a primordial ability to approximately estimate the number of items, or numerosity. To date it is debated whether numerosities presented in categorically different formats, that is as temporal sequences versus spatial arrays, are represented abstractly in the brain. To address this issue, we identified the behavioral characteristics and neuronal codes for sequential and simultaneous number formats in crows. We find a format-dependent representation by distinct groups of selective neurons during the sensory encoding stage. However, an abstract and format-independent numerosity code emerges once the encoding phase is completed and numerosities needed to be memorized. These results suggest a successive two-stage code for categorically different number formats and help to reconcile conflicting findings observed in psychophysics and brain imaging.

[1] Animal Physiology Unit, Institute of Neurobiology, University of Tübingen, Auf der Morgenstelle 28, 2872076 Tübingen, Germany. *email: andreas. nieder@uni-tuebingen.de

Counting is a sophisticated cognitive capability inextricably tied to symbolic competence and therefore unique to humans. However, our advanced numerical skills are built on a primordial and evolutionarily ancient ability to approximately estimate the abstract number of items in a collection, its numerosity[1,2]. Comparative research, both in the wild and in the laboratory, has shown that different species of animals from diverse zoological groups, such as mammals[3,4], birds[5,6], reptiles[7,8], amphibians[9,10], fish[11,12], and insects[13,14], can discriminate numerosity.

The neurobiological foundations of numerical competence have best been studied in nonhuman and human primates. Here, the endbrain, most notably the six-layered cerebral cortex, contains a dedicated brain network for processing numerical information[15]. Neurons in these brain areas selectively respond to the number of objects in arrays and show tuned activity to preferred numbers[16,17]. Birds similarly exhibit sophisticated numerical abilities[18–22], even though a radically different architecture emerged in their endbrains over 320 Mio years of parallel evolution[23–25]. Neurons in the crow endbrain area "nidopallium caudolaterale (NCL)", a proposed equivalent of the primate prefrontal cortex (PFC)[26,27], are activated by several cognitive factors[28–30]. NCL neurons also respond to the number of items that are displayed in spatial arrays and thus can be perceived from this specific, simultaneous number format[31–33].

However, numerosity as an abstract quantity must also be represented when items are presented one-by-one in temporal succession, but its underlying neuronal substrate is unclear. Both psychophysical[34–37] and brain imaging studies in humans[38] report conflicting results and, depending on the respective findings, argue that sequential and simultaneous number formats access the same approximate number system, or different systems, respectively. However, because psychophysics only measures the output of subjects, and brain imaging explores metabolic brain changes at relatively low temporal and spatial resolution[39], neither of these approaches alone is able to give a definitive answer. The only neuronal data available so far stem from the monkey parietal cortex and suggest an involvement of both format-dependent and -independent populations of neurons[40]. We therefore address the question of format-dependency in crows and set out to identify the neuronal codes for sequentially and simultaneously enumeration processes.

## Results

**Behavioral performance.** We trained two crows on a delayed match-to-sample task in which they had to assess the number of (one to four) dots in a sample phase, memorize it over a brief delay period, and match it to the number of dots shown in the test phase (Fig. 1). The sample numerosities were displayed either as sequentially presented single dots (sequential format; Fig. 1a) or as simultaneous-presented dot arrays (simultaneous format; Fig. 1d). Both presentation formats were alternated in a pseudo-random fashion from trial to trial. To ensure that the crows discriminated the number of dots rather than non-numerical stimulus features, we applied standard and control conditions for both number formats (see Methods for details). Controls in the sequential presentation format eliminated temporal factors that may co-vary with increasing numbers of sequential items, such as the total duration of the sample period, the duration of individual items and pauses in between, the total visual energy (or total area across time, respectively), and the regularity (rhythm) of the item sequence (Fig. 1b). For the simultaneous format, the overall area and density of the dots of variable size that appeared at random locations on the gray background circle was equalized in control conditions (Fig. 1e).

We first determined and compared the crows' performance to the sequential and simultaneous number formats, respectively, to explore behavioral evidence of a shared representational system. The crows' average performance for the sequential (crow B: $70.9 \pm 4.8\%$, crow J: $73.4 \pm 4.0\%$) and simultaneous format (crow B: $77.7 \pm 3.1\%$, crow J: $83.5 \pm 2.3\%$) was significantly above chance for sample numerosities and presentation formats (Binomial test, $p < 0.001$). Performance to control conditions was better than to standard conditions for both formats (two-tailed $t$-test, $p < 0.01$), possibly a reflection of better familiarity with standard stimuli. Importantly, the percent correct performances for all numerosities, formats, and conditions were significantly better than chance (binomial test, $p < 0.001$). The crows extracted the numerical information from the stimuli and did not use co-varying non-numerical features.

The behavioral performance functions to sequential (Fig. 1c) and simultaneous numerosities (Fig. 1f) illustrate two classic psychophysical phenomena in number discrimination: first, the numerical distance effect is evidenced by the finding that adjacent numerosities were more difficult to discriminate for the crows, which resulted in bell-shaped performance curves. Second, the numerical size effect is reflected by the finding that, at a given numerical distance, larger numerosities were more difficult to discriminate for the crows than smaller ones, which caused a progressive widening of the performance curves with an increase in sample numerosity. These classical signatures of numerosity discrimination were present for both number formats (Fig. 1g). The similarity of the performance curves may suggest that crows engage a format-independent approximate number system when judging sequential and simultaneous numerosity formats. We next explored the neuronal foundations underlying these behavioral numerosity representations.

**Single-neuron responses during the sample phase.** While the crows performed the task, we recorded the activity of 376 single neurons (crow B: 117 neurons, crow J: 259 neurons) in their endbrain area NCL[41] (crow B: 50 sessions, crow J: 76 sessions). Many neurons modulated their firing rate as a function of the number of items presented in the sample phase. Figure 2 depicts the response of four different neurons to the sequential format. Each neuron showed highest firing rates to one of the tested sample numerosities, such as one (Fig. 2a), two (Fig. 2d), three (Fig. 2g), or four (Fig. 2j). Importantly, the tuning of the four neurons was unaffected whether the standard or control protocols ("equal item duration") were presented (Fig. 2b, e, h, k), or how many items were shown in a row (Fig. 2c, f, i, l).

We tested the selectivity of neurons based on average firing rates in correct trials with a two-factor ANOVA, with main factors "number of items (one to four)" and "protocol (standard and control)", separately for the sequential and simultaneous format (criterion $p < 0.01$). Of the randomly recorded (i.e., not pre-selected) neurons, 17% (64/376 neurons) were numerosity selective for sequential numerosities (four of which shown in Fig. 2). A smaller proportion of 12% (45/376) of the neurons were numerosity selective for simultaneous numerosities.

Different selective neurons were tuned to different preferred numerosities in the two formats (Fig. 3a, b). We constructed average tuning curves by normalizing the individual tuning curves and pooling them according to preferred numerosity. Just as the behavioral performance functions, the average neuronal tuning functions showed a peak at their preferred numerosity and a progressive decline of activity with numerical distance from the preferred numerosity for both formats (Fig. 3c, d).

We wondered whether the numerosity-selective neurons might abstract across time and space during stimulus presentation to

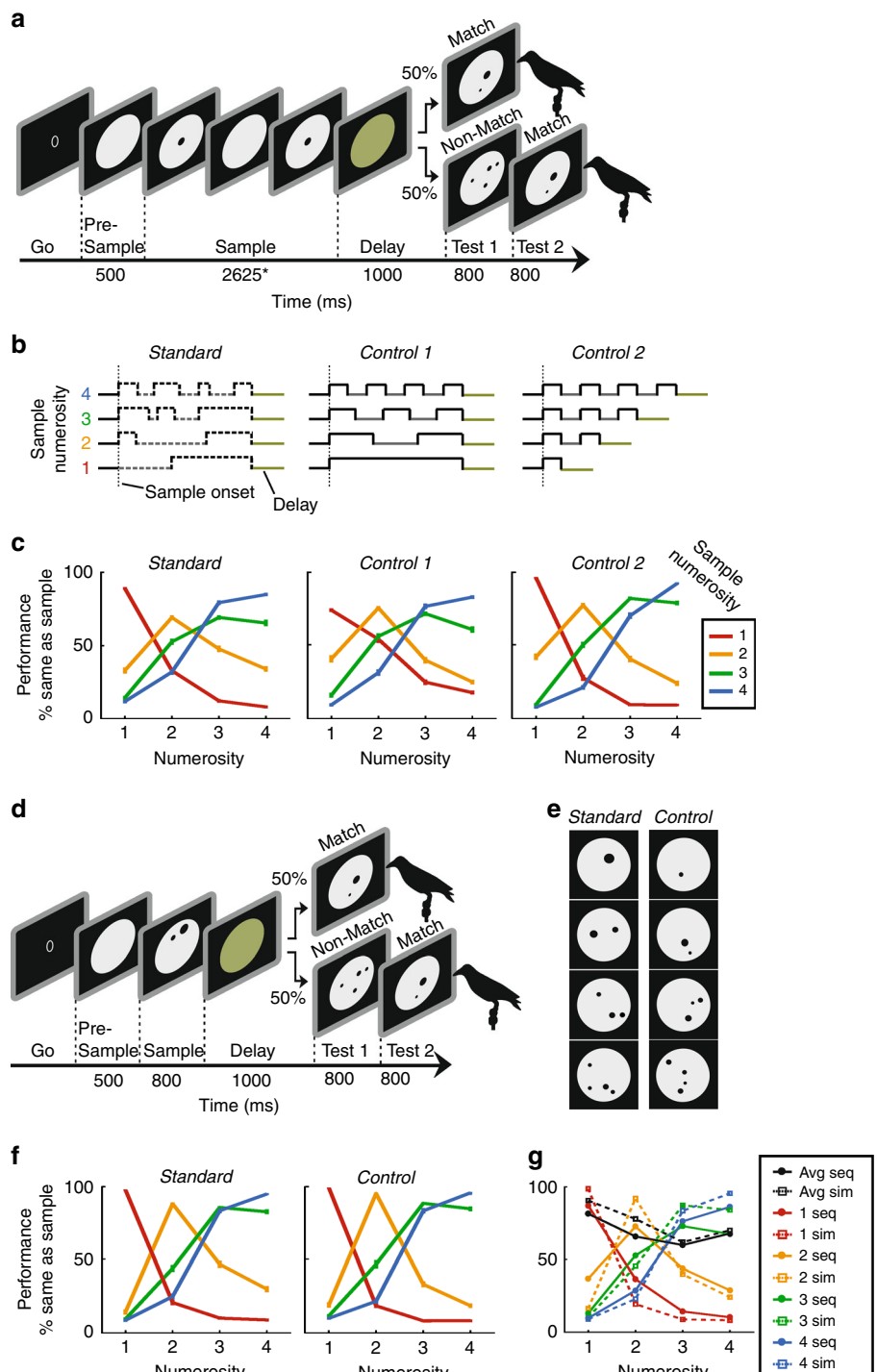

**Fig. 1 Task and behavioral performance. a** Sequential delayed match-to-numerosity task. A trial starts when the crow enters the light barrier. In this presentation format, the sample numerosity is presented one item at a time, interleaved by pauses. Match and Non-match trials occur with equal probability. The crow has to respond by leaving the light barrier, when sample and test numerosity are equal. **b** Temporal structure of standard and control trials in the sequential format followed by a 1-s delay. In standard trials, item and pause duration are randomly generated anew for each trial, illustrated by the dashed lines. As constraint, the entire sample duration must add to 2625 ms and item and pause duration must at least be 300 ms long. In temporal control 1 ("equal variance"), item and pause durations are kept constant within each numerosity but decrease across numerosity, with the overall sample duration kept constant. In control 2 ("equal item duration"), items and pauses have the same durations across numerosities, while the duration of the sample period increased with numerosity. **c** Behavioral performance separately for standard and control trials averaged over both crows (error bars ± SEM). The graphs illustrate the probability of a crow judging the test numerosity (x-axis) as being equal to the sample (color). **d** Simultaneous delayed match-to-numerosity task. Similar to **a**, except that sample dots appear at once for 800 ms. **e** Example standard and control stimuli for the simultaneous presentation format. The control encompasses equal area and equal density across numerosities. Standard stimuli appear as test displays in the sequential presentaion format as well. **f** Behavioral performance for standard and control trials averaged over both crows in the simultaneous presentation format; layout as in **c**. **g** Performance functions of both crows averaged across standard and control conditions for sequential (solid lines) and simultaneous presentation formats (dotted lines). Average percent correct performance per numerosity is depicted by black lines. All error bars indicate ± SEM.

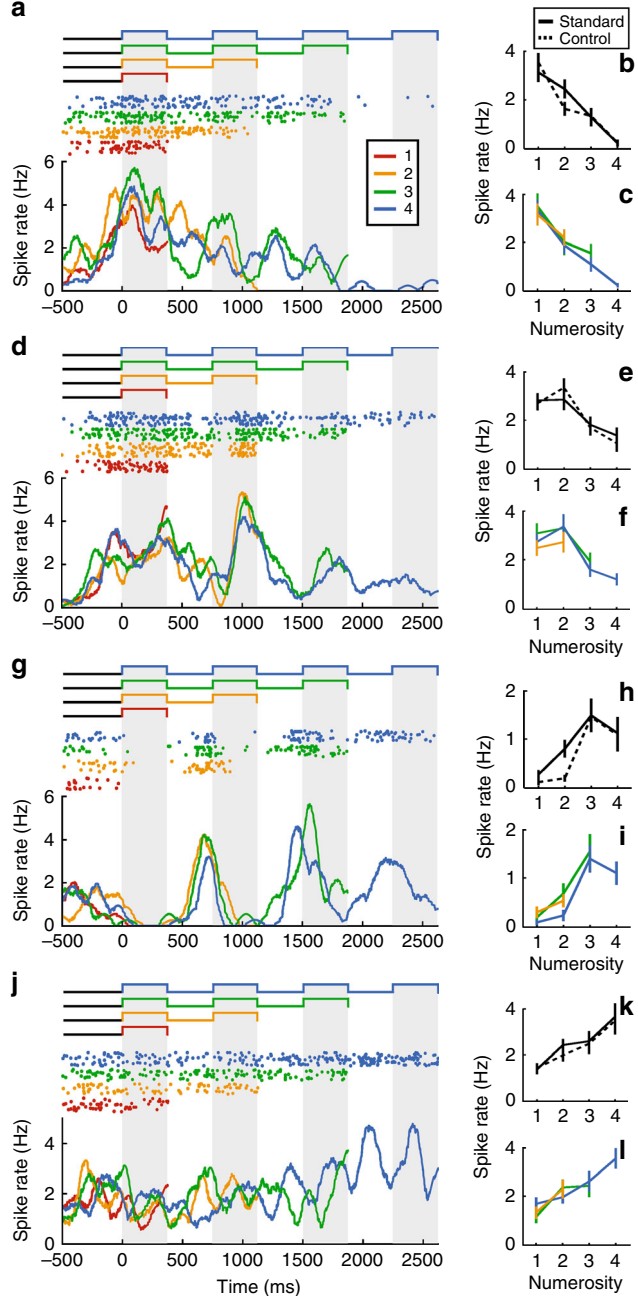

**Fig. 2 Neuronal responses to numerosity during the sample period. a–l** Single-neuron responses during sample presentation to the sequential presentation format. The three depicted example neurons preferred numerosities 1 (**a**), 2 (**d**), 3 (**g**), and 4 (**f**). Big panels: neuronal activity over the course of sample presentation during sequential control 2 trials. Top: dot-raster histogram, where dots represent action potentials and one row = one trial. Background shading indicates item present. Sample presentation started at time point 0 ms. Bottom: Averaged spike density function. **b, e, h, k** Average activity of the respective neurons during standard and control trials. **c, f, i, l** Average activity split by final item count. All error bars indicate ± SEM.

encode the same numerosity in both formats; alternatively, two specialized populations of neurons could encode the input number formats separately. We found the latter scenario realized: only 3.5% (13/376) of the sequentially and simultaneously tuned neurons were numerosity selective to both number formats, which was close to the calculated chance probability of 2.04%

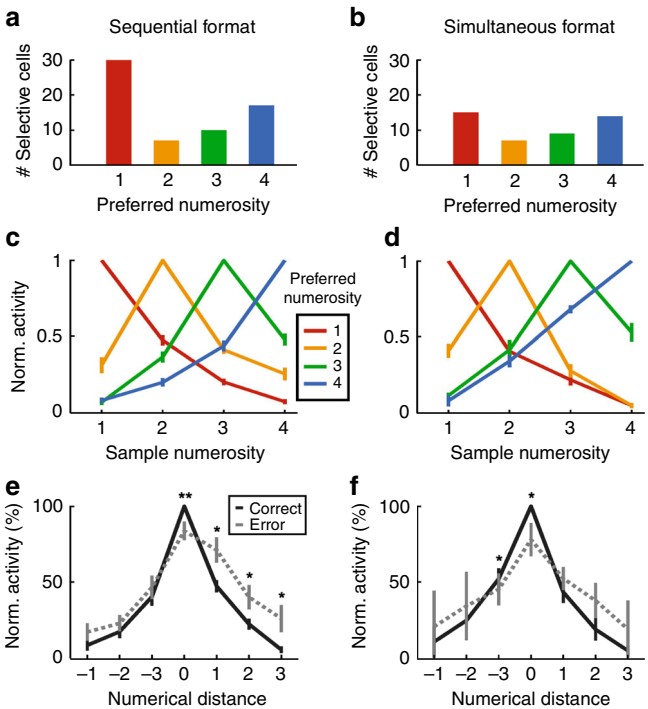

**Fig. 3 Tuning to numerosity during the sample period. a, b** Distribution of neurons that preferred each of the presented numerosities in both formats. **c, d** Normalized responses averaged over all neurons preferring the same numerosity. Tuning functions display peak response to preferred numerosities with a decrease for more distant numerosities. **e, f** Average tuning curves for correct compared to error trials. Neuronal responses in error trials are deteriorated, suggesting behavioral relevance *$p < 0.05$, **$p < 0.01$, Wilcoxon signed rank test). All error bars indicate ± SEM.

(Eq. (1)) (binomial test, $p = 0.047$). In addition, the neurons' preferred numerosities in the two number formats was not correlated ($r = 0.37$ (Spearman correlation, $p = 0.21$)), meaning that these 13 neurons did not prefer similar numerosities across formats. This suggests that two independent neuronal populations in the NCL were responsible to encode sequential and simultaneous numerosities during the sample phase.

These neurons were behaviorally meaningful, because their responses were distorted whenever the crows made judgment errors (Fig. 3e, f). The firing rates to the preferred numerosities, which was 100% by definition for correct trials, were significantly reduced in error trials (sequential format: 84%, $p = 0.006$, $n = 62$; simultaneous format: 78%, $p = 0.024$, $n = 38$, both Wilcoxon signed rank test). Thus, if the neurons failed to encode their preferred numerosities adequately, then the crows were at risk of making judgment errors.

**Single-neuron responses during the delay phase.** After sample presentation, the crows had to memorize the numerosity for 1 s until the test came up. We tested whether neurons were selective to numerosity during the delay irrespective of the number format. Indeed, 10% (37/376) of all recorded neurons were exclusively selective to numerosity during the delay (two-factor ANOVA, $p < 0.01$). Figure 4a–c displays three such exclusively numerosity-tuned delay neurons preferring numerosity 4, 2, and 1. The time course of neuronal firing shows that these neurons responded similarly to their respective numerosity independent of presentation format, and particularly towards the end of the delay phase (Fig. 4a–c). A negligible fraction of neurons (3.4%, 13/376

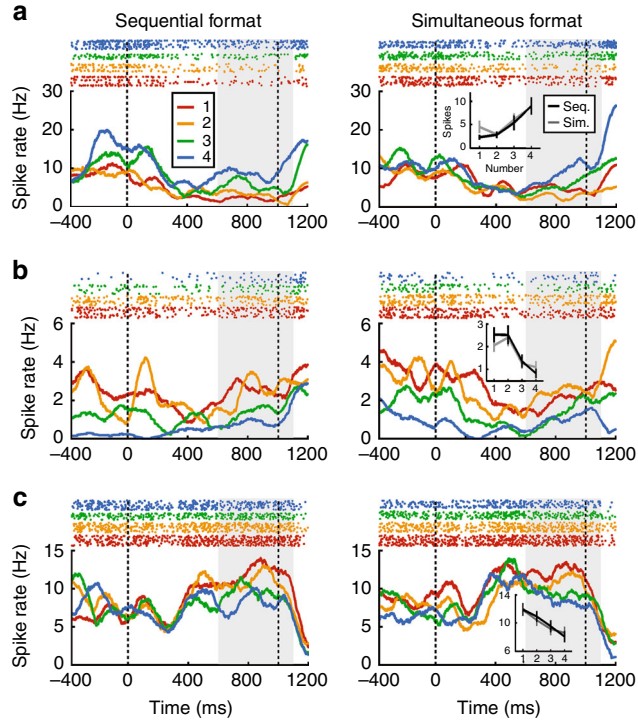

**Fig. 4 Neuronal responses during the delay.** Example responses of delay neurons shown separately for sequential (left column) and simultaneous presentation formats (right column). Example neurons prefer numerosity 4 (**a**), 2 (**b**), and 1 (**c**). Top panels: dot-raster histogram, bottom panels: averaged spike density functions. Dashed lines indicate delay on- and offset. Gray shaded area signifies the ANOVA analysis window. Insets: Average activity of the respective neuron to numerosity during both presentation formats (error bars indicate ± SEM).

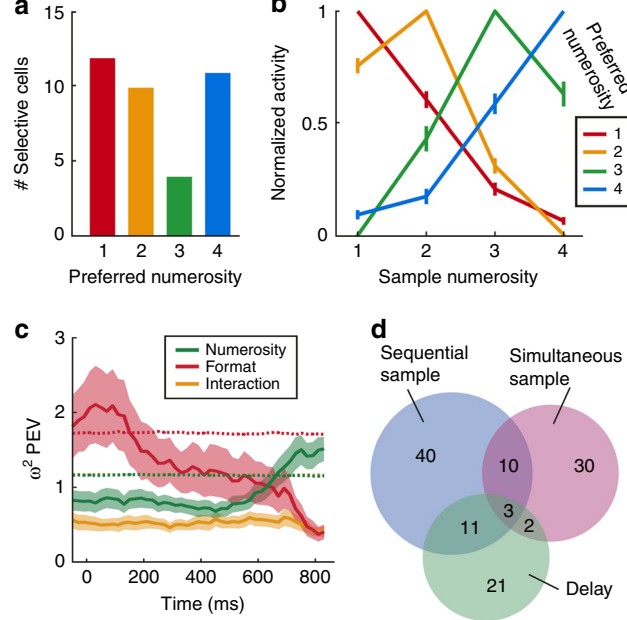

**Fig. 5 Characterization of neuronal responses during the delay.** **a** Distribution of neurons that preferred specific numerosities in both formats. **b** Average normalized tuning curves for all delay neurons preferring the same numerosity (error bars ± SEM). **c** Effect size ($\omega^2$ percent-explained variance) of numerosity and presentation format (sequential vs. simultaneous) on firing rates over time of numerosity-selective delay neurons ($n = 37$). Time point 0 ms is delay onset. Shading signifies ±SEM. Dashed lines indicate significance thresholds (95th percentile of shuffled distribution); the "interaction" threshold is superimposed by the "numerosity" threshold. **d** Venn diagram illustrating the distribution of numerosity-selective neurons across task phases. All error bars indicate ±SEM.

neurons) was selective to both numerosity and presentation format.

As can be seen from the frequency plot, we found only numerosity selective neurons preferring each of the four presented numerosity during the delay period (Fig. 5a). Average tuning functions of all numerosity-selective delay neurons show a characteristic peak at their preferred numerosity and a drop-off for more distant numerosities (Fig. 5b), again a reflection of a neuronal numerical distance effect.

The Venn diagram in Fig. 5d summarizes the counts of numerosity-selective neurons found in the sample and delay periods. The numbers in the blue and pink circles show the numbers of neurons that were exclusively selective to the sequential ($n = 64$) and simultaneous format ($n = 45$), respectively, during the sample period. Only 13 neurons were selective to both formats in the sample period, thus emphasizing the engagement of separate neuron populations during the sample phase. The cell counts in the green circle depict neurons that were exclusively numerosity selective to both formats during the delay period ($n = 37$). The overlap of the green circle with the sample circles means that some of these delay-selective neurons were also selective during the previous sample period. Thus, the blue and pink cell counts on the one hand, and the green cell count on the other hand, were derived from two different trial intervals.

A negligible proportion of the neuron tuned in the delay phase were also tuned to numerosity during the sample phases of the two number formats (Fig. 5d), thus emphasizing the engagement of separate neuron populations.

The histograms of the example neurons (Fig. 4a–c) suggest that delay-selective neurons switch between different types of

information they encode between the start and end of the delay phase. To quantify how much information about the sample numerosity and the format was conveyed by delay-selective neurons at any moment in the delay period, we performed a sliding-window percent-explained-variance analysis ($\omega^2$ explained variance; see Methods for details)). At the beginning of the delay period, the neurons primarily encode the format (Fig. 5c). Towards the end of the delay phase, format information strongly decreased and was replaced by numerosity information (Fig. 5c). Thus, just before the crows require information about the memorized numerosity to make a decision in the following test phase, numerosity became the dominant information carried by the neurons.

**Neuronal population coding.** So far, our single-neuron data suggest that one population of neurons, namely those cells that were exclusively numerosity selective to both formats during the delay period ($n = 37$), maintains numerical information in working memory independently from presentation formats. To explore to which extent these neuronal representations were indeed similar across formats, we applied a machine-learning algorithm using a linear multi-class support vector machine (SVM) as a classifier[42]. We first trained the classifier on firing rates recorded during the sequential number format and cross-validated the model with firing rates which were not used for training from the same presentation format. Then, we tested the classifier's accuracy in predicting numerosity with firing rates recorded during the simultaneous format, and vice versa. Cross-validation

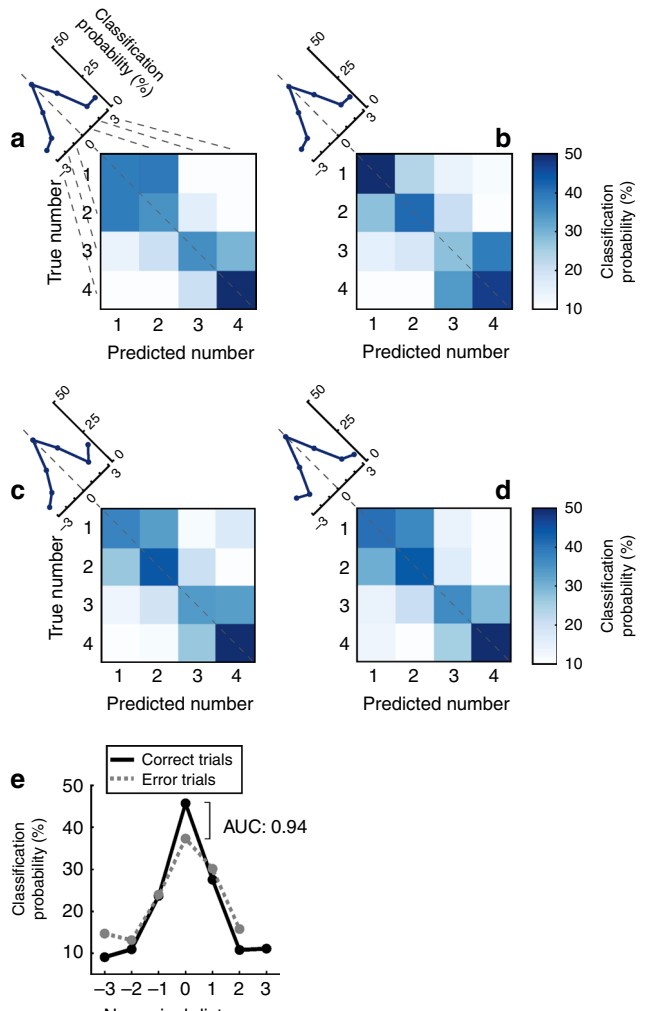

**Fig. 6 Population decoding during the delay based on the SVM classifier.**
**a**–**e** Decoding accuracy based on the SVM classifier. Confusion matrices
(**a**–**d**) derived from training classifiers on firing rates from the delay phase.
Rows indicate the true numerosity that the crow has seen, and the columns
the predicted numerosity. Tuning functions at the end of the confusion
matrix' diagonal depict average classification probability over all
numerosities. The confusion matrices in **a**–**d** differ in the data that were
used for training and testing. **a** Accuracy of the classifier trained and cross-
validated on delay firing rates after presentation of sequential numerosities.
**b** as in **a** but with a preceding simultaneous numerosity. **c** Accuracy of
the classifier trained with firing rates when the sample was presented
sequentially, then tested with firing rates from simultaneous trials. **d** Accuracy of the classifier trained with firing rates when the sample was
presented simultaneously, then tested with firing rates from sequential
trials. **e** Average classification probability when tested with correct and
error trials. Chance level is 25%.

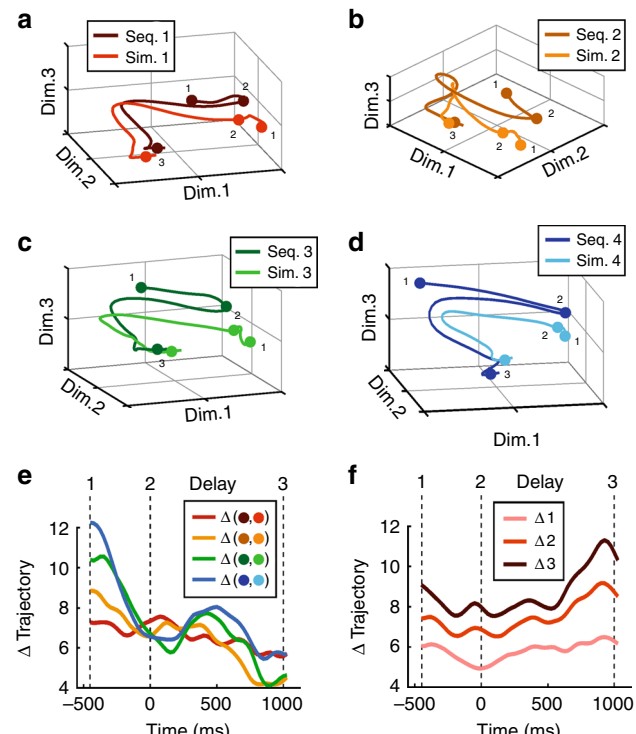

**Fig. 7 State-space analysis of delay response dynamics. a**–**d** State-space
trajectories for numerosities 1 (**a**), 2 (**b**), 3 (**c**), and 4 (**d**). Trajectories
depict the route separately for sequential and simultaneous presentation
formats. During onset of the last 500 ms in the sample period (1) and at
delay onset (2), responses to the two formats vary from each other. Over
the course of the delay, representations converge until the test comes up
(3). Euclidean distances between **e** sequential and simultaneous
trajectories separately for the four numerosities, and **f** between the
trajectories of the different sample numerosities. Distances were averaged
over pairs with the same numerical distance. Euclidean distances were
calculated over all 37 dimensions. Time point 0 ms is delay onset.

accuracies for both formats was comparable (Fig. 6a, b)
(sequential: 43.1% ± 5.5, standard deviation; simultaneous:
41.9% ± 4.3). The classifier trained on the sequential format pre-
dicted the correct numerosity in the simultaneous format with an
accuracy of 42.8% (±3.8) (Fig. 6c), and significantly above chance
level of 25%. Similarly, the classifier trained on the simultaneous
format predicted the correct numerosity in the sequential format
with an accuracy of 44.5% (±3.9) (Fig. 6d) (the 5th percentile
for both classifiers was above the 95th percentile of their respective
shuffled distribution). Based on the similar accuracies of

both cross-validations and both tests with the other presentation
format, we conclude that the classifier's performance was equally
well transferred from the sequential to the simultaneous, as from
the simultaneous to the sequential format. The average classifi-
cation probability functions show that classification errors mostly
occur for neighboring numerosities (Fig. 6a–d). However, mis-
classification probabilities decreased with increasing numerical
distance, a neuronal reflection of the behavioral numerical dis-
tance effect.

As a sign of the behavioral relevance of the neuronal
population, activity recorded during the crows' errors resulted
in more classification errors compared to correct trials (Fig. 6e).
Average classification performance in correct trials was 44.8%
(±2.9), whereas in error trials it was 37.1% (±3.7). This suggests
that neuronal discharges were behaviorally relevant and predicted
the crows' performance errors.

Finally, we analyzed the coding capacity and dynamics of the
population of numerosity-selective delay neurons (*n* = 37) by
performing a multi-dimensional state-space analysis (see Meth-
ods for details)[43]. Here, trajectories reflect the instantaneous
firing rates of the neuronal populations as they evolve over time.
For each preferred numerosity one to four, Fig. 7a–d displays the
trajectories for the sequential and simultaneous format. For all
numerosities, the trajectories representing sequential and simul-
taneous responses are spaced apart during the last 500 ms of the
sample phase. This indicates that, although numerosity is

represented during the last part of the sample period, the population also discriminates the formats and represents numerosity in a format-dependent way. However, during the course of the delay period, the trajectories converge and overlap, signifying that the population of neurons represented numerosity during working memory irrespective of the presentation format. This was quantified by measuring the Euclidean distances between the trajectories of the two number formats which converged for each numerosity (Fig. 7e). This indicates that information about the formats vanished in the delay period. Instead, the neuronal population discriminated better between the four numerosities, as demonstrated by increased Euclidean distances for increasing numerical distances (Fig. 7f).

## Discussion

Behavioral experiments in humans and monkeys have demonstrated that numerosity judgment show identical characteristics for sequential and simultaneous presentation formats[34,35]. This was taken as evidence that judgments of numerosity are abstract and based on the same approximate number system (but see also refs. [36,37] for contrasting findings). In the current study, we find equivalent performance signatures when crows assessed sequential and simultaneous numerosity formats, suggesting that crows also engage a format-independent approximate number system. The enumeration of sequentially presented items required the crows not only to enumerate (or add) numerosity but also to keep track of the serial position of the presented single item. A study in newborn chicks in which sequential and simultaneous numerosity discrimination during arithmetic tasks was compared similarly showed that these birds do not differentiate between the two formats[44].

Despite the same overall performance outcome, different neuronal mechanisms can give rise to behavioral equivalence. Whether number representations are format-dependent, format-independent, or a mixture of both, cannot be resolved from behavior alone. In fact, the neuronal data from crows argue for a neuronal two-stage process when these two fundamentally different number formats need to be represented. In each of these stages, different neuronal populations became engaged. First, during the sensory presentation stage, the number of sequentially presented items was extracted by one population of numerosity-tuned neurons, whereas the numerosity in dot arrays was represented by another population of numerosity-tuned neurons. These two numerosity-tuned populations showed strikingly similar neuronal coding characteristics but responded independently from each other to encode numerosity from the two formats. Both populations of neurons were behaviorally relevant and were predictive of erroneous trials for the respective number formats. At this sensory stage of number processing, neurons therefore responded format dependently.

However, once the encoding process was completed, a third population of numerosity-selective neurons became engaged. These delay-selective neurons represented numerosity irrespective of the presentation format. This third, format-independent population of neurons maintained numerical information in working memory and also predicted the crows' performance success. In summary, sequential and simultaneous number formats engage different and temporally succeeding populations of format-dependent and format-independent numerosity-selective neurons. These neuronal results are in agreement with previous findings in the convergently and anatomically distinctly evolved neocortex of primates. In macaque monkeys trained to perform almost identical sequential and simultaneous enumeration tasks, single-neuron recordings in the ventral intraparietal area (VIP) of the intraparietal sulcus (IPS) revealed the same two-step processing stages and coding mechanisms[40].

These findings in crows and monkeys help to explain seemingly equivocal results reported in human functional imaging. Several fMRI studies showed that arrays of dots, the most commonly used simultaneous number format, bilaterally activates areas of the IPS and prefrontal cortex in children[45,46] and adults[47–49]. However, when simultaneous and sequential number formats were directly compared in the same participants, a different activation pattern emerged for the sequential presentation: in contrast to activation patterns for dot arrays, activations were restricted to the right hemispheric IPS and the inferior frontal gyrus during processing of flashed dot sequences[38]. These findings argue that simultaneous and sequential formats are partly segregated in the brain because they engage different cortical networks. Importantly, however, these cortical networks overlapped in two brain areas, namely the right IPS and the right precentral/frontal gyrus. It seems that the format-dependent sensory encoding stage engages segregated format-dependent neural networks, whereas the coding stage following sensory enumeration activates the same format-independent neurons in IPS, thus giving rise to an abstract number representation later in the task. The brain engages both format-dependent and format-independent neuron populations to deal with simultaneous and sequential set sizes, but each during different and successive processing stages of a number task.

## Methods

**Subjects**. Two male hand-raised crows (Corvus corone corone) were trained on a match-to-numerosity task. Both crows were experienced with cognitive tasks from previous studies. Before the introduction of sequential numerosities, both crows proficiently judged simultaneous numerosity displays. The crows were housed in indoor aviaries in social groups. The crows were on a controlled feeding protocol during the training and recording period. Body weight was measured daily. The daily amount of food was given as reward during, or if necessary, after the sessions. Water was ad libitum available in the aviaries and during the experiments. All procedures were carried out according to the guidelines for animal experimentation and approved by the responsible national authorities, the Regierungspräsidium Tübingen, Germany.

**Experimental setup**. The experiment was conducted in a darkened operant conditioning chamber. The birds were perched in front of a touchscreen monitor (ART-Development PS-150, 15″, 60 Hz) that served for stimulus presentation. An automated feeder below the touchscreen delivered reward. The food reward consisted of food pellets and mealworms. A lamp on top of the feeder and speakers located behind the touchscreen provided additional feedback. An infrared light barrier activated by a reflector attached to the bird's head ensured a stable head position in front of the screen throughout the trial. We used the CORTEX system (National Institute of Mental Health) to carry out the experiment and collect behavioral data. Neuronal data were recorded using a PLEXON system (Plexon Inc., Dallas, Texas).

**Stimuli**. The crows had to assess whether displays in the sample and test periods contained the same number of one to four items. The displays consisted of black dots (diameter range: 2–4 mm) on a gray background circle (diameter: 3 cm). The sample numerosity was either presented sequentially by a sequence of centered single dots interleaved by pauses (sequential format) or simultaneously as dot arrays in which the dots were shown at random positions and with varying sizes (2–4 mm diameter) on the gray background circle (simultaneous format). Twelve different dot displays per numerosity were used, and all displays were replaced after each session with displays newly generated by computer software (custom written MATLAB script, MathWorks, R2016a). Furthermore, sample and test images, the latter always dot arrays, were never identical. Both measures ensured that the crows were paying attention to the number of dots, and prevented them from performing visual pattern matching.

One standard and two controls conditions in the sequential presentation format ensured that the crows could not use co-varying temporal cues to solve the task. The three conditions were applied to vary/control for (a) sample period duration, (b) individual item or pause duration, (c) rhythm, and (b) intensity over time. In the standard sequential format, numerosities were shown with randomly generated timings, but with the following constraints: the entire sample duration (overall time of dot presentation and pauses) had to add up to 2625 ms and the minimum time for an item or pause had to be 300 ms in duration. The item timings were generated anew for each trial. Thus, in the standard condition, sample period duration was constant, individual item or pause duration decreased with numerosity, rhythm was irregular, and intensity over time was variable. In addition, we implemented

two temporal control conditions that were alternately shown with the standard conditions: In the "equal variance" control conditions, sample period duration was constant (2625 ms), individual item or pause duration decreased with numerosity, rhythm was regular, and intensity over time decreased with numerosity. In the "equal item duration" control condition, sample period duration increased with numerosity, individual item or pause duration was constant across numerosities, rhythm was regular, and intensity over time increased with numerosity.

Similarly, standard and control conditions were applied in the simultaneous presentation format that prevented the crows from using co-varying visuo-spatial cues. In the standard conditions, non-overlapping dots of varying size were presented at varying locations on the gray background circle. In the control condition, the overall area of blackness by the dots as well as the density between the dots was equalized across numerosities. Density was calculated as the distances between the centers of the items. In contrast to our previous study[31], we did not include a control for total contour length, or circumference, because this visual parameter only had a negligible effect on NCL neurons. In addition, the large number of stimulus conditions forced us to focus on the most important non-numerical parameters in order to record enough stimulus and condition repetitions proper neuronal statistics. Standard and control simultaneous conditions were shown with equal proportions in each session.

The presentation formats (sequential and simultaneous) were displayed pseudo-randomly and balanced in each session. In addition, the sample numerosities (1–4) were shown in a balanced and pseudo-randomized fashion.

**Task**. The birds were trained on a delayed match-to-sample task with the numbers of dots as discriminanda. The crow initiated a trial by positioning its head facing the monitor whenever a go-stimulus (white circle, 2° visual angle) was shown, thus closing an infrared light barrier, and maintaining this position throughout the trial. To indicate that the light barrier had been entered, the bird heard a click sound and the go-stimulus vanished. Whenever a crow made premature head movements and thereby left the light barrier during an ongoing trial, this trial was terminated.

After the go-stimulus, the main task started with an empty background circle for 500 ms. Next, the sample numerosity was displayed. The sample numerosity was pseudo-randomly either presented sequentially or simultaneously. To signify the end of sample presentation, the subsequent delay period was indicated by a green background circle shown for 1000 ms. A test stimulus followed the delay. The dots of the test numerosity were always presented simultaneously (dot array) and the appearance was never identical to the sample. The first test was either the same or a different numerosity as the sample. If the test was the same numerosity (match), the crow had to answer by moving the head, thus leaving the light barrier. If the first test showed a different numerosity smaller or larger than the sample numerosity (non-match), the crow was not allowed to move its head but remain in the light barrier for 800 ms longer until a second test display appeared that was always a match, and the crow needed to respond. Match and non-match trials were shown pseudo-randomized with equal proportions. Correct trials yielded a food reward; incorrect trials yielded a time-out (8 s). Aborted trials were repeated at a later, pseudo-random time point.

To sum up, two sequential (standard and control) and two simultaneous presentation formats (standard and control) were shown in match and non-match trials in a balanced and pseudo-random fashion per session. In the sequential format, the two control conditions ("equal variance" and "equal item duration") alternated between sessions.

**Surgery and recordings**. The surgery was performed while the crow was under general anesthesia with a mixture of ketamine (50 mg/kg) and Rompun (5 mg/kg xylazine). The crow was placed in a stereotaxic holder. We targeted the medial part of NCL by performing a craniotomy at 5 mm anteriorposterior and 13 mm mediolateral on the right hemisphere. This part of NCL, termed mNCL, is known to contain highly associative neurons. Two manual microdrives containing four electrodes each (2 MΩ; Alpha Omega Co.) were implanted at the craniotomy. In addition, a miniature connector for the headstage and a small holder for attaching the reflector were implanted. Each recording session started with adjusting the electrodes until a proper neuronal signal was detected on at least one channel. The neurons were never pre-selected for any involvement in the task. Single-cell separation was done offline (Plexon Offline Sorter, version 2.8.8). No obvious anatomical organization of location preferences was detected.

**Data analysis**. All data analysis was carried out with MATLAB (MathWorks, R2016a). The neuronal population was defined as all single units that express a firing rate of at least 0.5 Hz and were present for at least three correct trials per condition (numerosity 1–4 × sequential/simultaneous × standard/control). A total of 376 neurons fulfilled these criteria and constitute the data base of this paper.

Numerosity selectivity in the sample phase was calculated separately for both presentation formats (sequential and simultaneous). A two-factor ANOVA with numerosity and standard/control as factors was used. Only neurons with a significant numerosity effect ($p < 0.01$) and no condition (standard/control) or interaction effect ($p > 0.01$) were called numerosity selective. The analysis windows for the ANOVA were shifted by 100 ms after stimulus onset (simultaneous presentation format) or item onset (sequential presentation format) to account for

neuronal response latencies and lasted for the entire stimulus or item presentation duration, but maximally up to 800 ms. In the delay phase, numerosity selectivity was determined by an ANOVA as well; the factors were numerosity and presentation format. The analysis window was set to 600 ms after delay onset and lasted for 500 ms. Standard numerosity 1 trials were omitted for the analysis, since these trials had a prolonged pre-sample phase to introduce some variance (only the firing rates of numerosities 2, 3, and 4 during the first dot presentation were used).

Tuning functions of numerosity-selective neurons were normalized to be between 0 (minimum firing rate; 0%) and 1 (maximum firing rate; 100%). All normalized tuning functions of individual neurons that had the same preferred numerosity were then averaged. To evaluate whether tuning properties (i.e. their preferred numerosity) were similar for neurons that were numerosity selective in both sample presentation formats, we tested the correlation of their preferred numerosity in both presentation formats with a Spearman's correlation coefficient.

The probability of how many neurons are expected to be numerosity selective by chance during both presentation formats in the sample phase was calculated with:

$$P(\text{seq} \cap \text{sim}) = P(\text{seq}) \times P(\text{sim}) = \frac{64}{376} \times \frac{45}{376} \approx 2.04\% \qquad (1)$$

with $P(\text{seq} \cap \text{sim})$ denoting the probability that a neuron is numerosity selective in the sample phase in both presentation formats, $P(\text{seq})$ the probability of being numerosity selective in the sequential presentation format and $P(\text{sim})$ for the simultaneous format.

Error analysis for the sample phase included all sample phase selective numerosity neurons that were present for at least three error trials for their preferred numerosity. In the sequential presentation format, this was the case for 62/64 neurons, in the simultaneous format for 38/45. Differences in firing rates between correct and error trials were tested with a Wilcoxon signed rank test.

The percent-explained variance (PEV) analysis served to quantify the information content about numerosity and presentation format over the time course of the delay. For each numerosity-selective delay neuron ($n = 37$), we used a sliding-window analysis (300 ms duration, 20 ms steps) and started 200 ms prior to delay onset. For each time window, we calculated $\omega^2$ with a two-factorial ANOVA to test the relationship between firing rates and stimulus category, and repeated this process with shuffled labels 1000 times (permutation test) to obtain a null distribution. The null distribution was obtained for each factor, namely numerosity, presentation format, and their interaction. To illustrate the explained variance for the entire population, we averaged the obtained $\omega^2$ values from each neuron and time window as well as the $\omega^2$ values from the null distributions. A given $\omega^2$ value was plotted at the time point from the middle of the sliding window. We determined the populations' information content to be significant, when the PEV value ($\omega^2$) was above the 95th percentile of the null distribution.

We also used multi-class SVMs. The aim of building and testing a decoder was to see how similar numerosities are represented across presentation formats. We included neurons that were numerosity selective in the delay phase, and were recorded for at least 32 correct trials and 1 error trial per numerosity and presentation format ($n = 20$ neurons). We excluded erroneous numerosity 1 trials from testing, since only few neurons had any error trials for this condition.

The multi-class SVM (LIBSVM toolbox for MATLAB, version 3.23) was trained with firing rates from one of the presentation formats (only correct trials). Default parameters from the toolbox delivered the best results, except that we used a polynomial kernel with one degree—which is linear—and used probability estimates for classification. Firing rates were averages taken 700 ms after delay onset for 400 ms. The model was cross-validated with the "leave-one-out" method, where one trial of every condition is left out for later testing and the other 31 trials are used to build the model. Before building, the firing rates were neuron-wise z-scored; cross-validation and test sets were normalized with $\mu$ and $\sigma$ obtained from z-scoring the training set. This process repeats 32 times, so that every trial serves as test trial once. After the cross-validation process, we build a model with all 32 trials per condition. This model is then tested with firing rates from the other presentation format in correct trials. The procedure was executed for both presentation formats; the model was built with sequential and tested with simultaneous trials, and vice versa. The depicted classification performances in confusion matrices show how likely the classifier predicts the true numerosity or confuses it with neighboring ones. Classification performance curves are averaged over iterations and presentation formats. This whole process of building, cross-validating, and testing repeats 100 times. In every iteration, new random trials for building and testing are drawn from each neuron per condition.

Finally, we wanted to see how the classifier performed on error trials, i.e. trials in which the crows responded wrongly. Therefore, we build a classifier with data from both presentation formats. Hence, for each numerosity, we used 32 sequential and 32 simultaneous trials from the 20 neurons to train the classifier. We applied a cross-validation just as before and then tested the model with error trials. For this testing, we randomly drew two error trials per numerosity (one error trial per presentation format) and had the model predict the shown numerosity. Since we only take two error trials per numerosity, but many neurons for which more than one error trial per numerosity and presentation format was recorded, we repeated the process of drawing two random error trial per condition 32 times. This served to get a more reliable error decoding performance. This process was repeated 100 times to account for the randomness of trials that were selected.

Differences in classification performances between correct (from cross-validation) and error trials were quantified with an AUC (area under the receiver operating curve). This measure is independent of the number of data points used. It gives a measure of how well two distributions are separable. A value of 0.5 means that the distributions are completely overlapping and therefore not separable; a 1 means that the distributions perfectly separable.

A population state-space analysis served to reveal population dynamics that might be hidden at the single unit level. At each point in time, the activity of $n$ recorded neurons is defined by a point in $n$-dimensional space, with each dimension representing the activity of a single neuron. Dimensionality reduction to the first three dimensions (that explain 48% of the neuronal covariance) using a principal component analysis results in trajectories that are traversed for different neuronal states, i.e. the four numerosities and the two number formats. These trajectories reflect the instantaneous firing rates of the respective neuronal population as they evolve over time.

For the population state-space analysis, we used all numerosity-selective delay neurons ($n = 37$). For each neuron, spike trains were averaged across trials over of the same condition (numerosity × presentation format), then smoothed with 150 ms Gaussian kernel, binned (300 ms, in 25 ms steps), and finally neuron-wise z-scored before calculating the principal components. Z-scoring prevents the state-space dynamics to be monopolized by only a few highly discriminative single neurons. The trajectories start 600 ms before delay onset and end 200 ms after delay offset. In the plots are the first three dimensions shown. To evaluate how population dynamics change over time in respect to presentation format, we calculated the Euclidean distances over the entire $n$-dimensional space (37-dimensional space) from the trajectories that represent the same numerosity in different presentation formats. Furthermore, to see how numerical information evolves during the course of the delay, we calculated the Euclidean distances between each pair of numerosities within the presentation formats (SEQ: 1v2, 1v3, 1v4, 2v3, 2v4, 3v4; SIM: 1v2, 1v3, 1v4, 2v3, 2v4, 3v4). Then we pooled the Euclidean distances from pairs with the same numerical distance across presentation format, e.g. the Δ2 trajectory is an average of the distances SEQ: 1v3 and 2v4, and SIM: 1v3 and 2v4.

**Reporting summary**. Further information on research design is available in the Nature Research Reporting Summary linked to this article.

## Data availability
The data that support the findings of this study are available from the corresponding author upon reasonable request.

## Code availability
The code that supports the findings of this study is available from the corresponding author upon reasonable request.

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

## Acknowledgements

This work was supported by a DFG grant NI 618/3-1 to A.N.

## Author contributions

A.N. and H.M.D. designed the study, interpreted the data, and wrote the manuscript. H.M.D. performed experiments and analyzed the data. A.N. supervised the study.

## Competing interests

The authors declare no competing interests.
