## [Peer Review File · Nature Communications]

Reviewers' Comments:

Reviewer #1:

Remarks to the Author:

Ditz and Nieder trained two crows to execute a numerosity-based delayed match-to-sample task under either a sequential or simultaneous format. They examined whether the two formats accessed the same, or different, approximate number systems. They found neurons that responded to specific numerosities and asked whether these neurons behaved in a format-dependent (fired during either sequential or simultaneous tasks) or format-independent (fired during both sequential and simultaneous tasks) manner.

To answer the above question the authors look at both the behavioral and neural data. With respect to the behavioral data, despite the fact that (as one would expect) the sequential format was harder than the simultaneous format, the two psychophysical functions generated by the two formats were very similar. First, both formats yielded a numerical distance effect where the more separated the numerosities the easier the discrimination (eg, 2 v 4 is an easier discrimination than 2 v 3). Second, both formats yielded a numerical size effect in that comparisons between higher numerosities are harder than comparisons between lower numerosities (1 v 2, which is separated by 1, is easier than 3 v 4, which is also separated by 1). Together, the behavioral data suggest that both formats have a shared (format-independent) representational system.

On the other hand in the sample phase it seems the neural data point to two populations of neurons coding the sequential and simultaneous number formats. In the delay phase, however, there seems to be a switch. Early in the delay the activity carried information about format, but later in the delay the activity was more geared towards numerosity.

Overall, there was a format-dependent neural code during encoding, a format-dependent neural code during the early stages of the delay, and a switch to a format-independent neural code during later parts of the delay. The overall switch reconciles the findings from numerous studies that argue one or the other mechanism.

A very nice paper with implications that cut across species. It should be published pending the answers below.

Main Issues

1. Figure 1: The caption mentions error bars but I could see no error bars in Figures 1C and 1F. This is important because one thing that does look a bit different between the sequential and simultaneous formats is the behavioral profile for numerosity 2: the sequential plot for 2 is much flatter than the simultaneous plot for 2. It would seem that the lower overall performance on the sequential task was driven by poorer overall performance with numerosity 2. So I need to know whether the 2 plot for the standard, control 1, and control 2 are significant in the sense of whether they show a quadratic trend (ie, a peak at 2). If the sequential 2 plots shows a quadratic trend then that would better convince me that the signatures for the numerosity discrimination were present in both formats, thus supporting the conclusion that the representational system, at least as derived from the behavioral data, are format-independent.

2. In a similar vein, the tuning curves in Figure 3, you've normalized them so that the highest value is represented by 1. Normalizing is good, but it seems like you've normalized them within each numerosity category. Would it not be better and more representative of the population of data, if you normalized them relative to the highest activity in your entire sample as opposed to the "1" plot normalized to the highest activity in the 1 sample, the "2" plot in the 2 sample, etc? If you normalized

them to the highest activity across all numerosities then I think the tuning curves in Figure 3 will more closely match the curves we saw in Figure 1C and 1F in that we could compare easier across the sequential and simultaneous formats. Same argument goes for Figure 5B.

3. I am struggling reading Figure 5, the pie chart. When you say that 37/376 neurons “were exclusively selective to numerosity during the delay (2-factor ANOVA)” am I correct in assuming that neither the factor of format nor the format x numerosity interaction was significant? So if there are 37 neurons exclusive to numerosity, and 11 overlap with sequential, 2 overlap with simultaneous, and 3 both sequential and simultaneous, that means that either a) there must be 21 and not 37 that are exclusively selective to numerosity (as per my definition) or b) my understanding of what you mean when you say “exclusive” is wrong. Please clarify. Likewise, this comment also relates to the first sentence under “Neuronal Population Coding” where the claim is made that one population of neurons maintain numerical information independent of format. Again, I read 11 dependent on the sequential format and 2 dependent on the simultaneous format.

4. Along a similar line, I found the labels used in Figure 5C to be very confusing with the text at the bottom of page 6. In the text you talk about “number format” which really is just “format” as well as the label “number”, but in the text you have the labels “number” and “format”. I suggest the labels in Figure 5C be “numerosity” and “format” and you refer to them as such in the text as well (rather than “number format”).

5. Finally, with respect to Figure 7, I just want to be clear about the interpretation. The way I read it is that during the delay numerosity is represented in working memory independent of format. The way it is written it makes it sound like during the last part of the sample phase only format is represented. Isn't it the case that numerosity is still represented in working memory but just in a format-dependent way?

Minor Issues

1. Page 4: please provide the actual p value for the nonsignificant “control condition was comparable to the standard conditions (one-tailed t-test, $p > 0.05$).”

2. Figure 1. I never wish to criticize someone for incorporating MORE controls in their experiment, but I was just wondering why if on “standard trials the item and pause duration are randomly generated anew for each trial” was there a need for the control 1 and control 2 conditions (other than, of course, if you didn't have it some reviewer would ask why). Was it so that you could produce consistent data to generate a “control” figure (ie, presumably just randomly generating the item and pause durations would have made it difficult to generate a “control” figure).

3. I just want to be clear that in the sequential format the stimuli presented sequentially was always “1”? In other words, did you ever present a “2” followed by a “1”? Here's why I ask. Is it possible that numerosity is being coded on the simultaneous task but position (first, second, third, fourth) coded on the sequential task? To untangle this possibility, then, another type of a control trial would present (on the sequential task) two dots in the first position.

4. What do you mean by “randomly recorded neurons”?

5. Please indicate the number of neurons obtained from each bird along with (as you have) the number of recording sessions.

6. Under Delay Phase, in that first paragraph, you reference Figure 3A-C twice but I think you mean to reference Figure 4A-C both times?

Reviewer #2:

Remarks to the Author:

The paper reports very interesting results showing different neuronal population coding for sequential and simultaneous format of non-symbolic numerosities, as well as the later emergence of a format-independent numerosity code in birds. I have some comments for improvement and/or clarification.

- P. 3 line 14: 'a functional equivalent': may I suggest a more cautionary form, like "an alleged equivalent" or similar? Whether the avian NCL correspond (functionally) or not to the mammalian prefrontal cortex is quite debated.
- P. 3 End of first para. The references quoted look a bit confusing for the non-specialist reader. For some species the authors reported quite specific papers (e.g. a paper on pigeon), for others very general reviews (fish); there is one mention to invertebrates (that are likely to possess very different neural substrates for number cognition) but no mention at all of the other major vertebrates groups, i.e. reptiles (e.g. Gazzola et al., (2018) *Biology Letters* <https://doi.org/10.1098/rsbl.2018.0649>) and amphibians (e.g. Stancher et al (2015). *Animal Cognition*, 18: 219-229). May I suggest to provide one or two examples for each major taxonomic group, possibly referring to different species in different groups (e.g. a couple of species for mammals, a couple of species for birds, reptiles... etc)? That would be more understandable for a generic reader.
- P. 3 Line 11: birds similarly exhibit sophisticated numerical abilities: please provide some evidence that it is so, with examples from different species.
- P. 4 line 13 from bottom: Why a one-tailed test?
- P- 4 Why there was no control for contour length? This need to be discussed because one can expect coding of boundaries quite relevant for the visual system.
- Discussion p. 9, first para. It is perhaps worth noting here that there is even a study in birds that compared sequential and simultaneous number discrimination during arithmetic tasks, also showing no difference at the behavioural level (Rugani et al. (2009). *Arithmetic in newborn chicks*. *Proceedings of the Royal Society of London B*, 276: 2451-2460).

Response to the referees

NCOMMS-19-28730

“Format-dependent and format-independent representation of sequential and simultaneous numerosity in the crow endbrain”

Please note that we mildly modified our title to make it more accessible for the general readership.

The comments of Reviewer #1 have been incorporated as follows:

Ditz and Nieder trained two crows to execute a numerosity-based delayed match-to-sample task under either a sequential or simultaneous format. They examined whether the two formats accessed the same, or different, approximate number systems. They found neurons that responded to specific numerosities and asked whether these neurons behaved in a format-dependent (fired during either sequential or simultaneous tasks) or format-independent (fired during both sequential and simultaneous tasks) manner.

To answer the above question the authors look at both the behavioral and neural data. With respect to the behavioral data, despite the fact that (as one would expect) the sequential format was harder than the simultaneous format, the two psychophysical functions generated by the two formats were very similar. First, both formats yielded a numerical distance effect where the more separated the numerosities the easier the discrimination (eg, 2 v 4 is an easier discrimination than 2 v 3). Second, both formats yielded a numerical size effect in that comparisons between higher numerosities are harder than comparisons between lower numerosities (1 v 2, which is separated by 1, is easier than 3 v 4, which is also separated by 1). Together, the behavioral data suggest that both formats have a shared (format-independent) representational system.

On the other hand in the sample phase it seems the neural data point to two populations of neurons coding the sequential and simultaneous number formats. In the delay phase, however, there seems to be a switch. Early in the delay the activity carried information about format, but later in the delay the activity was more geared towards numerosity.

Overall, there was a format-dependent neural code during encoding, a format-dependent neural code during the early stages of the delay, and a switch to a format-independent neural code during later parts of the delay. The overall switch reconciles the findings from numerous studies that argue one or the other mechanism.

A very nice paper with implications that cut across species. It should be published pending the answers below.

Main Issues

1. Figure 1: The caption mentions error bars but I could see no error bars in Figures 1C and 1F. This is important because one thing that does look a bit different between the sequential and simultaneous formats is the behavioral profile for numerosity 2: the sequential plot for 2 is much flatter than the simultaneous plot for 2. It would seem that the lower overall performance on the

sequential task was driven by poorer overall performance with numerosity 2. So I need to know whether the 2 plot for the standard, control 1, and control 2 are significant in the sense of whether they show a quadratic trend (ie, a peak at 2). If the sequential 2 plots shows a quadratic trend then that would better convince me that the signatures for the numerosity discrimination were present in both formats, thus supporting the conclusion that the representational system, at least as derived from the behavioral data, are format-independent.

We are sorry for this misunderstanding. The error bars in Figure 1C and 1F represent the standard error of the mean. Since we averaged over 126 session and therefore divided the standard deviation by the square root of 126, the error bars are quite small but present. To better show that the classic signature of numerosity discrimination were present for all formats and numerosities, we incorporated **novel Fig. 1G**. This figure now shows the average peak performance functions (averaged across standard and control conditions) for sequential and simultaneous formats. We are also plotting the average percent correct performance per numerosity (black lines in Fig. 1G) and statistically tested percent correct performance per numerosity. The percent correct performances for all numerosities (including numerosity 2 in the sequential format), formats and conditions were significantly better than chance (Binomial test, $p < 0.001$). We are reporting this in the **second paragraph on page 4**.

2. In a similar vein, the tuning curves in Figure 3, you've normalized them so that the highest value is represented by 1. Normalizing is good, but it seems like you've normalized them within each numerosity category. Would it not be better and more representative of the population of data, if you normalized them relative to the highest activity in your entire sample as opposed to the "1" plot normalized to the highest activity in the 1 sample, the "2" plot in the 2 sample, etc? If you normalized them to the highest activity across all numerosities then I think the tuning curves in Figure 3 will more closely match the curves we saw in Figure 1C and 1F in that we could compared easier across the sequential and simultaneous formats. Same argument goes for Figure 5B.

Normalizing tuning functions is always a tricky aspect. Each of the different methods of normalizing firing rates has its advantages and disadvantages. We normalized the firing rates so that the discharge of each single unit to the preferred numerosity is set to 100%, and to the least preferred 0%. After that, all tuning functions for the same numerosity were averaged. We clarified this on **page 14, second paragraph**.

If we would normalize the functions to the highest activities across all numerosities, the tuning curves would be heavily influenced by neurons with high firing rates, and high firing rates occur arbitrarily for any preferred numerosity. In other words, neurons tuned to 1 do not necessarily have the highest firing rate, so a correlation of the height of the neuronal functions with the behavioral function, as expected by the reviewer, can not be expected. Moreover, we have been using the current normalization method (0 to 100%) for almost two decades for human, monkey, crow and network data. For the sake of comparability, we would strongly prefer to maintain our current normalization method.

3. I am struggling reading Figure 5, the pie chart. When you say that 37/376 neurons "were exclusively selective to numerosity during the delay (2-factor ANOVA)" am I correct in assuming that neither the factor of format nor the format x numerosity interaction was significant? So if there are 37 neurons exclusive to numerosity, and 11 overlap with sequential, 2 overlap with simultaneous, and 3 both sequential and simultaneous, that means that either a) there must be 21 and not 37 that are exclusively selective to numerosity (as per my definition) or b)my

understanding of what you mean when you say “exclusive” is wrong. Please clarify. Likewise, this comment also relates to the first sentence under “Neuronal Population Coding” where the claim is made that one population of neurons maintain numerical information independent of format. Again, I read 11 dependent on the sequential format and 2 dependent on the simultaneous format.

We are sorry for causing confusion. We have not been clear enough about the fact that the Venn-diagram combines two different trial phases, the sample and the delay phases.

The blue circle gives the number of neurons selectively tuned during the sample phase in the sequential format (40+10+3+11), which adds up to the 64 cells mentioned on **page 5, third paragraph** (“Of the randomly recorded neurons, 17% (64/376 neurons) were numerosity selective for sequential numerosities (four of which shown in Fig. 2). This is within-format testing. The pink circle provides the number of neuron (30+10+3+2) selective in the sample phase for the simultaneous format (“A smaller proportion of 12% (45/376) of the neurons were numerosity selective for simultaneous numerosities.” **Page 5, third paragraph**). This is also within-format testing.

The green circle gives the number of selective neurons during the delay period, which adds up to 37 neurons (21+11+3+2) („Indeed, 10% (37/376) of all recorded neurons were exclusively selective to numerosity during the delay (2-factor ANOVA, $p < 0.01$).” **page 6, third paragraph**). This is cross-format testing.

The overlap of the green circle with the sample circles means that some of these delay-selective neurons were also selective during the previous sample period. Thus, the blue and pink cell counts on the one hand, and the green cell count on the other hand, were derived from two different trial intervals. We clarified this now on **page 6, last paragraph**. We also extended the first sentence under “Neuronal Population Coding” on **page 7, third paragraph**, for clarity.

4. Along a similar line, I found the labels used in Figure 5C to be very confusing with the text at the bottom of page 6. In the text you talk about “number format” which really is just “format” as well as the label “number”, but in the text you have the labels “number” and “format”. I suggest the labels in Figure 5C be “numerosity” and “format” and you refer to them as such in the text as well (rather than “number format”).

Thank you for bringing this point to our attention. We changed the label in Figure 5C accordingly. Also, we changed the terminology in the main text as suggested by the reviewer (to be found now on **page 7, second paragraph**)

5. Finally, with respect to Figure 7, I just want to be clear about the interpretation. The way I read it is that during the delay numerosity is represented in working memory independent of format. The way it is written it makes it sound like during the last part of the sample phase only format is represented. Isn't it the case that numerosity is still represented in working memory but just in a format-dependent way?

The reviewer is correct, numerosity is represented in the last part of the sample period but just in a format-dependent way. We clarified this on **page 8, final paragraph**.

Minor Issues

1. Page 4: please provide the actual p value for the nonsignificant “control condition was comparable to the standard conditions (one-tailed t-test, $p > 0.05$).”

Thank you for this comment. As suggested, we now changed statistics to a two-tailed t-test separately for sequential and simultaneous trials. For both presentation protocols, the crows performed better in control than in standard trials. This is now mentioned on **page 4, second paragraph**. Despite this difference, the percent correct performances for all numerosities, formats and conditions were significantly better than chance.

2. Figure 1. I never wish to criticize someone for incorporating MORE controls in their experiment, but I was just wondering why if on “standard trials the item and pause duration are randomly generated anew for each trial” was there a need for the control 1 and control 2 conditions (other than, of course, if you didn’t have it some reviewer would ask why). Was it so that you could produce consistent data to generate a “control” figure (ie, presumably just randomly generating the item and pause durations would have made it difficult to generate a “control” figure).

We apologize that our description concerning the three sequential conditions was not clear enough.

The three conditions were applied to vary/control for a) sample period duration, b) individual item or pause duration, c) rhythm, and b) intensity over time. In the standard condition, sample period duration was constant, individual item or pause duration decreased with numerosity, rhythm was irregular, and intensity over time was variable. In the ‘equal variance’ control conditions, sample period duration was constant, individual item or pause duration decreased with numerosity, rhythm was regular, and intensity over time decreased with numerosity. In the ‘equal item duration’ control condition, sample period duration increased with numerosity, individual item or pause duration was constant across numerosities, rhythm was regular, and intensity over time increased with numerosity.

We clarified the need for these conditions in the results on **page 4, first paragraph**, and in the methods on **page 12, second paragraph**.

3. I just want to be clear that in the sequential format the stimuli presented sequentially was always “1”? In other words, did you ever present a “2” followed by a “1”? Here’s why I ask. Is it possible that numerosity is being coded on the simultaneous task but position (first, second, third, fourth) coded on the sequential task? To untangle this possibility, then, another type of a control trial would present (on the sequential task) two dots in the first position.

The reviewer is correct that the task required the crows to also keep track of the serial position of the individual items. We are now discussing this on **page 9, first paragraph**.

4. What do you mean by “randomly recorded neurons”?

Randomly recorded neurons are neurons that have not been pre-selected based on response criteria to obtain an unbiased estimate of the proportion of selective neurons. We clarified this on **page 5, third paragraph**.

5. Please indicate the number of neurons obtained from each bird along with (as you have) the number of recording sessions.

We recorded 117 neurons from crow B, and 259 neurons from crow J. We now added this information on **page 5, second paragraph**.

6. Under Delay Phase, in that first paragraph, you reference Figure 3A-C twice but I think you mean to reference Figure 4A-C both times?

Yes, we meant Figure 4. Thank you, this is now corrected.

The comments of Reviewer #2 have been incorporated as follows:

The paper reports very interesting results showing different neuronal population coding for sequential and simultaneous format of non-symbolic numerosities, as well as the later emergence of a format-independent numerosity code in birds. I have some comments for improvement and/or clarification.

- P. 3 line 14: 'a functional equivalent': may I suggest a more cautionary form, like "an alleged equivalent" or similar? Whether the avian NCL correspond (functionally) or not to the mammalian prefrontal cortex is quite debated.

We agree and followed the reviewer's suggestion by using the term "proposed equivalent" on page 3, second paragraph.

- P. 3 End of first para. The references quoted look a bit confusing for the non-specialist reader. For some species the authors reported quite specific papers (e.g. a paper on pigeon), for others very general reviews (fish); there is one mention to invertebrates (that are likely to possess very different neural substrates for number cognition) but no mention at all of the other major vertebrates groups, i.e. reptiles (e.g. Gazzola et al., (2018) *Biology Letters* <https://doi.org/10.1098/rsbl.2018.0649>) and amphibians (e.g. Stancher et al (2015). *Animal Cognition*, 18: 219-229). May I suggest to provide one or two examples for each major taxonomic group, possibly referring to different species in different groups (e.g. a couple of species for mammals, a couple of species for birds, reptiles... etc)? That would be more understandable for a generic reader.

We thank the reviewer for this recommendation, and we are happy to incorporate more references. We are now citing two original research papers for each of the taxonomic groups mentioned (mammals, birds, reptiles, amphibians, fish, and insects), including the ones suggested by the reviewer. This modification can be found on page 3, first paragraph.

- P. 3 Line 11: birds similarly exhibit sophisticated numerical abilities: please provide some evidence that it is so, with examples from different species.

We gladly added several references from different species demonstrating numerical abilities in birds in the second paragraph of page 3.

- P. 4 line 13 from bottom: Why a one-tailed test?

This comment was also raised by reviewer #1 (first minor comment). As suggested, we now changed statistics to a two-tailed t-test separately for sequential and simultaneous trials. For both presentation protocols, the crows performed better in control than in standard trials. This is now mentioned on page 4, second paragraph. Despite this difference, the percent correct performances for all numerosities, formats and conditions were significantly better than chance.

- P- 4 Why there was no control for contour length? This need to be discussed because one can expect coding of boundaries quite relevant for the visual system.

Contour length, or circumference, is indeed a non-numerical parameter that has to be taken into account. We therefore had incorporated such a circumference control (total circumference of all

items equated for all stimuli in a trial) in a previous study in crows (Ditz and Nieder, 2015, PNAS). In this study, we saw that neurons were only very rarely sensitive to this parameter. In the current study, the large number of stimulus conditions forced us to focus on the most important non-numerical parameters, because we were already struggling to record enough stimulus and condition repetitions in order to perform proper cellular statistics. Even more controls would have jeopardized the experiment. We are discussing these aspects now on **page 12, third paragraph**.

- Discussion p. 9, first para. It is perhaps worth noting here that there is even a study in birds that compared sequential and simultaneous number discrimination during arithmetic tasks, also showing no difference at the behavioural level (Rugani et al. (2009). Arithmetic in newborn chicks. Proceedings of the Royal Society of London B, 276: 2451-2460).

Thank you for this suggestion. We are now discussing this elegant study at the end of the **first paragraph, page 9**.

Reviewers' Comments:

Reviewer #1:

Remarks to the Author:

The authors have done a good job of responding to my queries and I am happy to endorse the manuscript.

Reviewer #2:

Remarks to the Author:

I think the authors have addressed adequately all my concerns. I believe this paper deserves to be published.